

# Serum albumin and albuminuria predict the progression of chronic kidney disease in patients with newly diagnosed type 2 diabetes: a retrospective study

Yujiao Li[1,*], Xiaobing Ji[2,*], Wenji Ni[1], Yong Luo[1], Bo Ding[1], Jianhua Ma[1] and Jian Zhu[1]

[1] Department of Endocrinology, Nanjing First Hospital, Nanjing Medical University, Nanjing, Jiangsu, China
[2] Department of Nephrology, Nanjing First Hospital, Nanjing Medical University, Nanjing, Jiangsu, China
* These authors contributed equally to this work.

Corresponding author
Jian Zhu, drzhujian@hotmail.com

## ABSTRACT

**Background:** Diabetes-related kidney disease is associated with end-stage renal disease and a high mortality rate. However, data on risk factors associated with kidney disease in patients with newly diagnosed type 2 diabetes mellitus (DM) remains insufficient. The aim of the present study was to identify the risk factors significantly associated with chronic kidney disease progression in patients with newly diagnosed type 2 DM.

**Methods:** We reviewed a total of 254 consecutive patients who were newly diagnosed with type 2 diabetes at Nanjing First Hospital from January to December 2014. They were observed for two years, and baseline and biochemical variables were used to identify significant predictors of kidney failure progression. Kidney failure progression was defined as a $\geq$ 30% increase in serum creatine level.

**Results:** The mean age of patients was 58.96 years, 37.4% were women, and 57.1% had hypertension. Kidney function progressed in 40 patients (15.75%). Multivariable logistic regression analyses showed that serum albumin ($p = 0.015$) and microalbuminuria ($p < 0.001$) were associated with kidney failure progression in patients with newly diagnosed type 2 DM. Those with lower estimated glomerular filtration rate (eGFR; 30–60 ml/min/1.73 m$^2$) at baseline had lower serum albumin levels compared to those of patients with higher eGFR. The albuminuria levels were higher in patients with lower eGFR than in those with eGFR $\geq$ 90 ml/min/1.73 m$^2$. Receiver operating characteristic curve analysis showed that the area under the curve was 0.754 (95% CI [0.670–0.0.837]).

**Conclusions:** The overall rate of chronic kidney disease progression is relatively high, and low serum albumin and high albuminuria levels are associated with kidney failure progression in newly diagnosed diabetic patients.

## INTRODUCTION

Diabetes-related chronic kidney disease (CKD) has become more common than CKD related to glomerulonephritis in both the general population and a hospitalized urban population in China (*Zhang et al., 2016*). Diabetic patients with CKD are at risk for progression of kidney disease to end-stage renal disease (ESRD), cardiovascular events, and mortality (*Retnakaran et al., 2006*; *Anders et al., 2018*). To prevent CKD progression, intensive glycaemic control, lifestyle modifications, and administration of renin-angiotensin system inhibitors are used for diabetic patients. However, the rate of CKD progression to ESRD has remained unchanged over the past two decades (*William, Hogan & Batlle, 2005*; *Anders et al., 2018*), with approximately 20 out of 10,000 adults with diabetes mellitus (DM) developing ESRD per year. Those numerous complications, comorbidities, and medications, as well as renal replacement therapy associated with advanced CKD and ESRD impose an enormous economic burden on the healthcare system (*D'Onofrio et al., 2017*; *Gerber et al., 2018*; *Duan et al., 2019*). Therefore, there is a need for epidemiologic studies to identify potentially modifiable risk factors to prevent the progression of kidney disease.

DM has approached epidemic proportions over the past decade worldwide. In China, the prevalence of diabetes has sharply increased. In a 2013 survey, 4% of adults have been previously diagnosed with diabetes (*Hu & Jia, 2018*); 6.9% of patients have received a new diagnosis according to the American Diabetes Association 2010 criteria. Several studies had identified some clinical markers predicting CKD progression in previously diagnosed diabetic patients (*Retnakaran et al., 2006*; *Dunkler et al., 2015*; *Go et al., 2018*; *Zhang, Ye & Pan, 2019*). However, data on the risk factors related to kidney function decline in newly diagnosed type 2 DM remain insufficient. Therefore, we undertook this retrospective observational study to examine the risk factors in this patient population.

## MATERIALS AND METHODS

### Study population

We conducted a retrospective analysis on the clinical data of patients who were newly diagnosed with type 2 DM in Nanjing First Hospital from January 2014 to December 2014. Patients were excluded if they had the following: (1) acute complications of DM on admission, such as diabetic ketoacidosis; (2) severe infectious diseases on admission, fever, urinary tract infection, haematuria, glomerulonephritis; (3) severe cardiovascular diseases, such as stroke, myocardial infarction, coronary artery bypass grafting, percutaneous coronary intervention, and heart failure; (4) severely impaired liver function; (5) psychiatric disorders or were pregnant or planning to conceive; (6) cognitive disorders, alcoholism, or history of drug abuse; (7) acute kidney injury upon admission and/or at the end of follow-up. Patients lost to follow-up or with incomplete data were also excluded. The patients ranged in age from 18 to 86 years. All patients had the same medical team for treatment and care. The study protocol was approved by the Institutional Review Board at Nanjing First Hospital, Nanjing Medical University, which waived the requirement for written informed consent from patients.

## Data collection

Data from all participants were collected anonymously. Type 2 DM was diagnosed according to the American Diabetes Association 2010 criteria (*Tankova, 2007*). Biochemical measurements were performed at the clinical laboratory at Nanjing First Hospital using a Beckman AU5800 clinical chemistry analyzer (Beckman Coulter, California, United States). Estimated glomerular filtration rate (eGFR) was calculated for each patient using the Modification of Diet in Renal Disease Study equation (*Ma et al., 2006*): eGFR (ml/min/1.73 m$^2$) = 175 × SCr-1.234*age-0.179 * 0.79 (if female). Albuminuria levels were measured at the same laboratory by a chemiluminescent method using the IMMULITE 2000XPi immunoassay system (Shimadzu, Kyoto, Japan). The results were divided into quartiles (>300, 100–300, 30–100, and <30 mg/24 h) and assigned a value (4, 3, 2, 1). The diagnosis of acute kidney injury was based on a serum creatine level increase of ≥0.3 mg/dL (26.52 μmol/L) or ≥1.5- to twofold from baseline level and a urine output of <0.5 ml/kg/h for more than 6 h.

## Outcomes

Kidney function decline was defined as a ≥ 30% increase in serum creatinine level. Patients whose serum creatine levels were increased because of acute kidney injury were excluded.

## Statistical analysis

Data were analysed with PASW Statistics18 software (SPSS (Hong Kong) Ltd., Hong Kong, China). Continuous variables were expressed as mean ± standard error of the mean. Statistical significance of differences between groups were determined by t-test or one-way analysis of variance. Ordinal data were expressed as M (Q1, Q3), and a Wilcoxon rank sum test was used for comparisons between the two groups. Proportions were compared using the chi-squared test and Fisher's exact test when the numbers were small. All variables were first analysed using univariate analysis. Variables with a $p$-value < 0.1 in the univariate analysis were then entered into a multivariable logistic regression analysis model to determine their net effects on kidney function decline. Odds ratios and their 95% confidence intervals (CIs) were used to assess the independent contribution of prognostic factors. A $p$-value < 0.05 was considered statistically significant.

# RESULTS

## Demographic and clinical characteristics of patients

The incidence of kidney function decline was 15.75% ($n$ = 40). The baseline demographics and clinical characteristics of the patients are listed in Table 1. The mean age of the progression and non-progression group were 62.90 ± 16.21 and 58.22 ± 14.33 years, respectively, and 62.6% of the patients were male ($n$ = 159). The mean body mass index of all enrolled was 25.21 ± 3.53, and 57.1% ($n$ = 145) of them had a history of hypertension. Kidney function decline was more likely among patients with a history of hypertension and a larger hipline measurement.

**Table 1 Baseline characteristics of individuals.**

|  | Non-Progression (n = 214) | Progression (n = 40) | Total (n = 254) | p-value |
|---|---|---|---|---|
| Sex (male) | 139 | 20 | 159 | 0.07 |
| Age (years) | 58.22 ± 14.33 | 62.90 ± 16.21 | 58.96 ± 14.70 | 0.06 |
| BMI (Kg/m$^2$) | 25.12 ± 3.34 | 25.68 ± 4.46 | 25.21 ± 3.53 | 0.35 |
| Waistline (cm) | 88.42 ± 8.70 | 90.03 ± 9.85 | 88.67 ± 8.89 | 0.30 |
| Hipline (cm) | 94.01 ± 7.52 | 96.48 ± 9.14 | 94.40 ± 7.83 | 0.06 |
| Systolic pressure (mmHg) | 130.20 ± 14.91 | 138.50 ± 17.50 | 131.50 ± 15.60 | <0.01 |
| Diastolic pressure (mmHg) | 78.73 ± 9.74 | 82.28 ± 10.86 | 79.29 ± 9.98 | 0.04 |
| Heart rate (beat/min) | 77.77 ± 7.71 | 75.65 ± 8.62 | 77.44 ± 7.87 | 0.12 |
| Comorbidities |  |  |  |  |
| Hypertension | 114 | 31 | 145 | <0.01 |
| Smoking | 53 | 9 | 62 | 0.80 |
| Drinking | 25 | 6 | 31 | 0.56 |
| Coronary heart disease | 43 | 7 | 50 | 0.71 |
| Diabetic retinopathy | 55 | 14 | 69 | 0.23 |
| Use of RAAS inhibitor | 102 | 25 | 127 | 0.09 |

**Note:**
RAAS: Renin-Angiotensin-Aldosterone system.

## Biochemical variables of the patients

The biochemical variables of the patients are listed in Table 2. The progression group had significantly lower haemoglobin, total protein, albumin, and indirect bilirubin levels in blood; it also had significantly higher glucose, triglyceride, low-density lipoprotein, and glycosylated haemoglobin A1c levels.

## Potential risk factors for CKD progression in the multivariate logistic regression

The multivariable logistic regression analysis was used to identify the risk factors for kidney function decline, and the results are listed in Table 3. After adjusting for confounders, both serum albumin and albuminuria were associated with kidney function decline in patients with newly diagnosed type 2 DM.

## Associations between serum albumin, albuminuria and eGFR

Those with lower eGFR level (30–60 ml/min/1.73 m$^2$) at baseline had lower serum albumin compared to those of patients with an eGFR 60–90 ml/min/1.73 m$^2$ and eGFR ≥ 90 ml/min/1.73 m$^2$ (Fig. 1). The albuminuria levels were higher in patients with lower eGFR level than in those with eGFR ≥ 90 ml/min/1.73 m$^2$ (Fig. 2).

## Receiver operating characteristic (ROC) curve estimation

An ROC curve was created by plotting sensitivity against specificity at various threshold settings. The x-axis represents 1-specifity, and the y-axis represents sensitivity, the model was well calibrated, and the area under the ROC curve is shown in Fig. 3 and Table 4.

**Table 2 Biochemical variables of the individuals.**

| | Non-Progression (*n* = 214) | Progression (*n* = 40) | Total (*n* = 254) | *p*-value |
|---|---|---|---|---|
| Hb (g/L) | 134.98 ± 16.67 | 127.95 ± 13.99 | 133.87 ± 16.46 | 0.01 |
| TSH (mIU/L) | 2.11 ± 1.88 | 1.88 ± 2.13 | 2.08 ± 1.92 | 0.48 |
| FT3 (pmol/L) | 4.37 ± 1.14 | 4.11 ± 0.49 | 4.33 ± 1.07 | 0.15 |
| FT4 (pmol/L) | 15.95 ± 5.84 | 15.56 ± 1.95 | 15.89 ± 5.42 | 0.68 |
| Total protein (g/L) | 68.83 ± 6.56 | 66.39 ± 4.73 | 68.44 ± 6.36 | 0.03 |
| Albumin (g/L) | 41.13 ± 3.61 | 38.92 ± 2.88 | 40.77 ± 3.59 | <0.01 |
| Direct bilirubin (μmol/L) | 4.05 ± 2.11 | 3.96 ± 1.99 | 4.03 ± 2.09 | 0.80 |
| Indirect bilirubin (μmol/L) | 8.67 ± 4.60 | 7.26 ± 3.85 | 8.45 ± 4.51 | 0.07 |
| Creatine (mol/L) | 82.66 ± 29.78 | 80.75 ± 38.38 | 82.20 ± 31.26 | 0.72 |
| Uric acid (μmol/L) | 340.83 ± 108.58 | 343.98 ± 102.87 | 342.16 ± 107.51 | 0.90 |
| FBG (mmol/L) | 11.58 ± 7.53 | 13.93 ± 7.67 | 11.94 ± 7.60 | 0.07 |
| HbA1c (%) | 8.74 ± 2.12 | 9.77 ± 2.21 | 8.90 ± 2.16 | <0.01 |
| Total cholesterol (mmol/L) | 4.84 ± 1.03 | 5.06 ± 1.28 | 4.88 ± 1.07 | 0.23 |
| Triglyceride (mmol/L) | 2.19 ± 2.18 | 3.13 ± 4.12 | 2.34 ± 2.59 | 0.03 |
| HDL (mmol/L) | 1.31 ± 0.35 | 1.33 ± 0.31 | 1.31 ± 0.34 | 0.66 |
| LDL (mmol/L) | 2.65 ± 0.63 | 2.91 ± 1.02 | 2.69 ± 0.71 | 0.03 |
| Albuminuria (mg/h) | 56.73 ± 141.89 | 282.10 ± 394.84 | 92.22 ± 218.52 | <0.01 |

Note:
Hb, haemoglobin; TSH, thyroid stimulating hormone; FT3, free triiodothyronine; FT4, free thyroxine; FBG, fasting blood glucose; HDL, high-density lipoprotein; LDL, low-density lipoprotein; HbA1c, glycosylated haemoglobin A1c.

**Table 3 Results of the multivariable analysis to determine the factors for kidney function decline.**

| Variable | Coefficient | Standard error | Statistic | *p*-value | OR (95% CI) |
|---|---|---|---|---|---|
| Albumin | −0.123 | 0.051 | 5.872 | 0.015 | 0.884 [0.801–0.977] |
| Albuminuria | 0.003 | 0.001 | 15.053 | <0.001 | 2.18 [1.558–3.050] |

The area under the ROC curve of albumin was 0.702 (95% CI [0.662, 0.782]), that of albuminuria was 0.708 (95% CI [0.608, 0.808]), and that of albumin combined with albuminuria was 0.754 (95% CI [0.670, 0.837]).

# DISCUSSION

The present study was undertaken to investigate the risk factors associated with kidney function decline in newly diagnosed type 2 DM patients. A few studies had examined the risk factors for CKD progression in diabetic patients (*Retnakaran et al., 2006*; *Dunkler et al., 2015*; *Shih et al., 2020*). However, all patients were diagnosed with DM before enrolment in these studies; therefore, the results could interfere with the use of medicine and lifestyle modifications, which makes those studies prone to bias. Early predictions and delay of disease progression are of great concern in the clinical management of patients with type 2 DM. Our study demonstrated the potential risk factors that need to be managed in newly diagnosed type 2 DM patients.

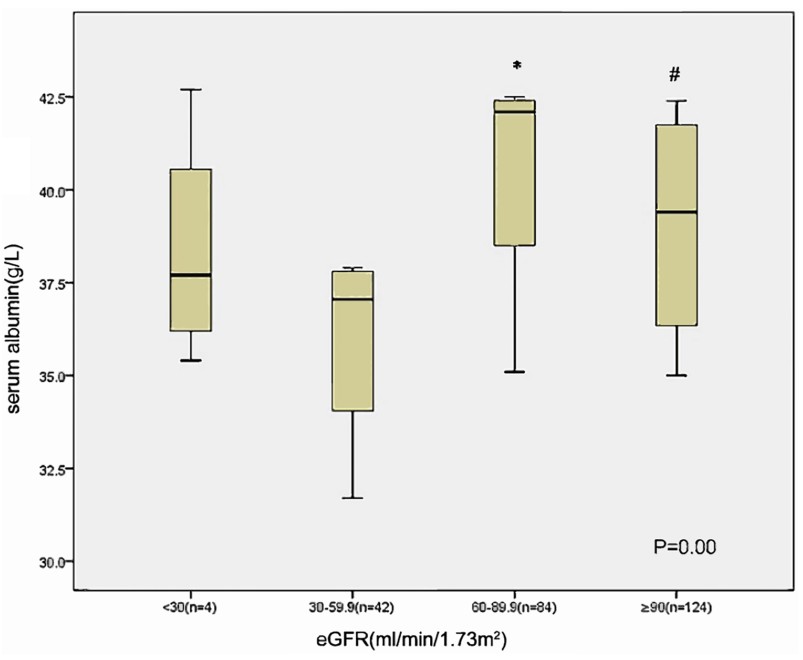

**Figure 1 Relationship between serum albumin and eGFR category in 254 patients at baseline.**
$^*p < 0.05$ eGFR 30–59 VS eGFR 60–89.9, #$p < 0.05$ eGFR 30–59 VS eGFR ≥ 90.

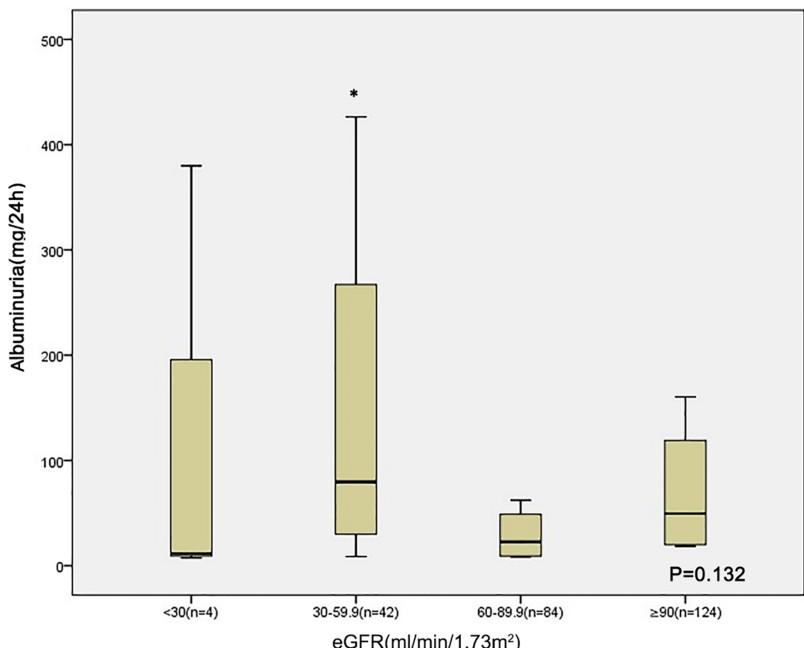

**Figure 2 Relationship between albuminuria and eGFR category in 254 patients at baseline.** $^*p < 0.05$
eGFR 30–59 VS eGFR ≥ 90.

In the present study, we observed an association between albuminuria and kidney function decline in type 2 DM patients, and the predictive power of albuminuria for progressive kidney disease has previously been shown in diabetic patients

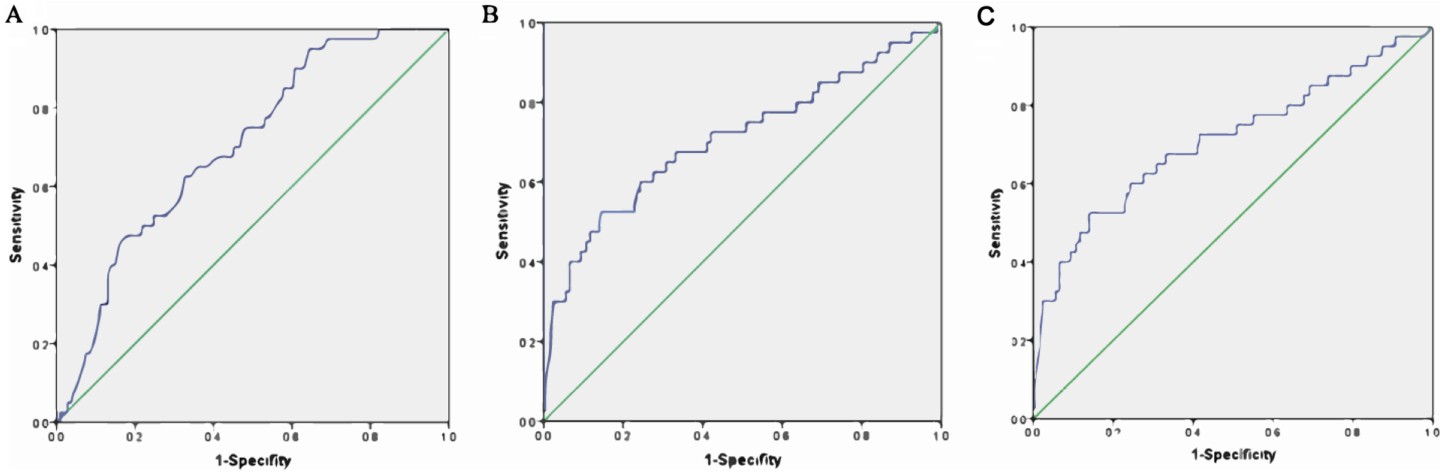

**Figure 3  ROC curve estimation.** (A) The AUC of albumin was 0.702 (−0.662, 0.782), (B) the AUC of albuminuria was 0.708 (0.608, 0.808), and (C) the AUC of albumin combined with albuminuria was 0.754 (95% CI [0.670–0. 0.837]).

**Table 4  ROC curve estimation.**

| Variable | AUC | Standard error | OR (95% CI) | p value |
|---|---|---|---|---|
| Albumin | 0.702 | 0.041 | 0.702 [0.662–0.782] | <0.001 |
| Albuminuria | 0.708 | 0.051 | 0.708 [0.608–0.808] | <0.001 |
| Combined | 0.754 | 0.051 | 0.754 [0.670–0.837] | <0.001 |

(*Spoelstra-de Man et al., 2001*; *Araki et al., 2008*). Several studies have found that many patients already have structural changes in kidney biopsy after the development of albuminuria (*Looker, Mauer & Nelson, 2018*; *Looker et al., 2019*), suggesting that albuminuria is a marker of established nephropathy rather than being a predictor of diabetic nephropathy. In our study, the albuminuria level is higher in the eGFR 30–60 group than in the eGFR ≥ 90 group at baseline; the small sample size may have contributed to the lack of significant difference between patients in the eGFR 60–90 and eGFR ≥ 90 groups. A recent study also showed that, in American Indians with type 2 DM, urine albumin to creatinine ratio reflects the progression of early structural glomerular lesions rather than early GFR decline (*Looker et al., 2019*). It is generally acknowledged that progression of albuminuria may well represent a progression of early vascular disease, eventually leading to a cardiovascular end point. Thus, it cannot sufficiently emphasize the importance of albuminuria in diabetic patients.

We found that serum albumin was associated with kidney function decline in newly diagnosed diabetic patients independent of albuminuria. Previous studies have shown that low albumin levels are strongly associated with cardiovascular disease, heart failure, and mortality in vulnerable populations (*Akirov et al., 2017*; *Gotsman et al., 2019*). Limited research has linked low serum albumin to kidney function decline (*Lang et al., 2014*; *Jiang et al., 2020*). Few studies have demonstrated that serum albumin is an important component of a multi-marker predictive model for progression to ESRD (*Keane et al.,*

*2006; Lang et al., 2018*). In our study, the serum albumin level was lower in the eGFR 30–60 group than in the eGFR ≥ 90 and eGFR 60–90 groups at baseline. A recent study showed that lower serum albumin levels are strongly and independently associated with the decline in kidney function in elderly individuals (*Lang et al., 2018*). Low albumin level could reflect an impairment in nutritional status or, in a blood dilution, be a marker that shows kidney damage or the progression of kidney failure. The potential mechanisms underlying these associations are unclear.

Only a few studies have so far reported that other risk factors independently predict CKD progression in diabetic patients. *De Cosmo et al. (2015)* found that mild hyperuricemia is strongly associated with the risk of CKD in patients with type 2 diabetes. Other risk factors include but are not limited to eGFR and hypertension (*Retnakaran et al., 2006*; *Dunkler et al., 2015*; *Novak et al., 2016*). We did not confirm this association, which may be explained by the difference in the race of patients enrolled.

Our study has several limitations. First, the study was a retrospective, single-centre study, thus making it prone to bias. Second, there was sample size inequality between the groups, and accordingly, the observed significant differences may have been due to the small number of patients in the progression group.

## CONCLUSION

Low serum albumin and high albuminuria levels are strongly associated with CKD progression in patients with newly diagnosed type 2 DM. Additional prospective studies in humans are required to confirm this relationship and to understand the underlying mechanisms.

## ACKNOWLEDGEMENTS

The authors would like to thank the patients enrolled in this study.

### Funding
The study and article processing charges were supported by the National Key R&D Program of China (No. 2018YFC1314100) and the Nanjing Medical Science Fund for Distinguished Young Scholars (JQX12006). The funders had no role in study design, data collection and analysis, decision to publish, or preparation of the manuscript.

### Grant Disclosures
The following grant information was disclosed by the authors:
National Key R&D Program of China: 2018YFC1314100.
Nanjing Medical Science Fund for Distinguished Young Scholars: JQX12006.

### Competing Interests
The authors declare that they have no competing interests.

## Author Contributions

- Yujiao Li performed the experiments, analyzed the data, prepared figures and/or tables, authored or reviewed drafts of the paper, and approved the final draft.
- Xiaobing Ji performed the experiments, analyzed the data, prepared figures and/or tables, authored or reviewed drafts of the paper, and approved the final draft.
- Wenji Ni performed the experiments, prepared figures and/or tables, and approved the final draft.
- Yong Luo performed the experiments, prepared figures and/or tables, and approved the final draft.
- Bo Ding performed the experiments, prepared figures and/or tables, and approved the final draft.
- Jianhua Ma conceived and designed the experiments, authored or reviewed drafts of the paper, and approved the final draft.
- Jian Zhu conceived and designed the experiments, analyzed the data, prepared figures and/or tables, authored or reviewed drafts of the paper, and approved the final draft.

## Ethics

The following information was supplied relating to ethical approvals (i.e., approving body and any reference numbers):

The study protocol was approved by the Institutional Review Board at Nanjing First Hospital, Nanjing Medical University, which waived the requirement for written informed consent from patients.

## Data Availability

The raw measurements are available in the Supplemental Files.

## Supplemental Information

Supplemental information for this article can be found online at http://dx.doi.org/10.7717/peerj.11735#supplemental-information.

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
