# Peer review of "Serum albumin and albuminuria predict the progression of chronic kidney disease in patients with newly diagnosed type 2 diabetes: a retrospective study"

_PeerJ, doi:10.7717/peerj.11735_

## Round 0.1 · original submission · Major Revisions

The authors should address the issues raised by reviewers 2 and 3. In particular, they should integrate clinical data, try to emphasise novelty of their study, and speculate about the mechanisms linking ipoalbuminemia to the clinical context described. English should be thoroughly revised.

Reviewer 1 ·

Basic reporting

Authors demonstrated that low serum albumin and high microalbuminuria levels are strongly associated with CKD progression in diabetic patients. The paper is written in a good english and is conform to professional standards. The literature references are sufficient and fits with the field knowledge.

Experimental design

no comment

Validity of the findings

I think the paper is not so original as the relationship between initial damage from exposition to hyperglycemia, the occurrence of microalbuminuria and progression to CKD is already well demonstrated in literature. These paper confirms these findings.

Reviewer 2 ·

Basic reporting

Major comments:
It is my impression that the terms renal function decline and CKD progression are used interchangeably throughout the manuscript.This is quite tricky and confusing as a decline in eGFR does not necessarily translate into progression in CKD.Also not all patients had CKD at baseline.Authors need to clarify and amend accordingly.Also the phrase ‘’kidney progression’’ in lines 27,31,38,111 is not correct and should be replaced.
Minor comments :
1.In line 82 replace word creatine with creatinine
2.In line 98 authors need to substitute phrase ‘’..mean age of progression and non-progression’’ with ‘’mean age of progressors and non-progressors’’
3.In line 156 authors need to supplement the word type in front of ‘’..2 diabetes’’.

Experimental design

Major comments:
1.There is a clear need to study natural course of T2DM and its complications in various ethnic populations and discover biomarkers that help predict development of these complications so as to be able to intervene early.There is also a paucity of data on the above with regards to exclusively newly diagnosed patients with T2DM.Authors of the present study are trying to address these issues by studying the change in renal function of a relatively large cohort of Chinese people with newly diagnosed T2DM.Although the follow up duration is not long a 30% change of eGFR from baseline allows for the study to capture early changes in renal function that could help predict adverse renal outcomes eg progression to CKD/ESRD.
2.The authors state that the primary endpoint of the study is a more than 30% decline in eGFR.This has been shown to consist a valid endpoint in the study of renal function decline in diabetes and the general population (Levey et al Am J Kidney Dis 2016).Since serum creatinine levels are subject to significant biological and preanalytic variability the authors should confirm whether this level of decline is sustained or is it just a one-off measurement at the end of follow up?
3.What is the proportion of people in the study on RAAS blockade given the known renoprotective effect of these agents(Lewis et al N Eng J Med 2001)?
4.What is the proportion of people with diabetic retinopathy (DR) in the study population?DR often coexists with kidney disease in diabetes(Afkarian et al JAMA 2016) and could also serve as a predictor of renal function decline in this population potentially even at an early stage(although it is most commonly seen with long-standing disease).

Validity of the findings

Major comments :
1.Have the authors looked into the diagnostics of multivariable logistic regression analysis model so as to confirm that their model is robust and generalizable?
Minor comments :
1.I would prefer the term multivariable compared to multivariate regression analysis.

Reviewer 3 ·

Basic reporting

English revision is needed

Experimental design

Retrospective design is a relevant limit

Validity of the findings

The novelty is limited

Additional comments

The manuscript by Jian Zhu et al. describes a vaste retrospective study conducted on newly diagnosed type 2 mellitus diabetes patients. The study aimed to investigate risk factors associated with CKD progression. Authors found macroalbuminuria and low serum albumin as factors associated with kidney function decline.
The following issues can be raised:
1. The result on microalbuminuria does not add any new knowledge , since this is a widely accepted dogma in nephrology. Conversely, some novelty can be found in the association between low albuminemia and CKD progression. Low albuminemia could reflect in an impairment of nutritional status or in a blood diluition. Both conditions would be however present in aneddotic cases and are generally associated with AKI, that represents an exclusion criterion in this study. The underlaying pathogenic mechanisms still remain to be explained.

2. Authors missed to explain how their results would impact on clinical practice.

3. The English language quality deserve a revision by a English native speaker (eg the expression wisely used by the authors 'kidney progression' sounds inappropriate, giving the idea of an improvement rather than a worsening. Authors might substitute it with'kidney failure progression'...)

---

## Round 0.2 · Minor Revisions

The manuscript has substantially improved, although there are still minor issues to be addressed.

Reviewer 2 ·

Basic reporting

Language of the manuscript has improved.It reads better now.
Again there seems to be a misconception regarding microalbuminuria.There is no substance microalbumin as such but microalbuminuria as a defined level of excretion of albumin in the urine as compared to normo- and macroalbuminuria.Authors should address this in the manuscript.

Experimental design

My comments on experimental design have been addressed.

Validity of the findings

Further analyses as outlined in my review have been performed.Comments sufficiently addressed.

Reviewer 3 ·

Basic reporting

Overall the manuscript is well presented and English language quality has been improved

Experimental design

Study design is appropriate and statistical analysis was complete

Validity of the findings

Novelty remains a bit limited, however the authors better underlined the utility of their findings and the current version appears more convincing.

Additional comments

The current version of the manuscript has been improved by the authors in accomplishment with my suggesions and comments.

However, I would like to suggest enclosure of 'Quality of life worsening' within ESRD outcomes in the introductory paragraph (please refer to and cite: D'Onofrio G. et al. Quality of life, clinical outcome, personality and coping in chronic hemodialysis patients. Ren Fail. 2017 Nov;39(1):45-53. doi: 10.1080/0886022X.2016.1244077)

---

## Round 0.3 · accepted · Accept

The authors have adequately addressed all the issues raised by the reviewers.